# Four-Dimensional Superimposition Techniques to Compose Dental Dynamic Virtual Patients: A Systematic Review

**DOI:** 10.3390/jfb14010033

**Published:** 2023-01-06

**Authors:** Ying Yuan, Qian Liu, Shuo Yang, Wulin He

**Affiliations:** 1Department of Orthodontics, Stomatological Hospital, Southern Medical University, Guangzhou 510280, China; 2Center of Oral Implantology, Stomatological Hospital, Southern Medical University, Guangzhou 510280, China

**Keywords:** digital dentistry, computer-assisted design (CAD), patient simulation, image fusion, diagnosis, oral, evidence-based dentistry

## Abstract

Four-dimensional virtual patient is a simulation model integrating multiple dynamic data. This study aimed to review the techniques in virtual four-dimensional dental patients. Searches up to November 2022 were performed using the PubMed, Web of Science, and Cochrane Library databases. The studies included were based on the superimposition of two or more digital information types involving at least one dynamic technique. Methodological assessment of the risk of bias was performed according to the Joanna Briggs Institute Critical Appraisal Checklist. Methods, programs, information, registration techniques, applications, outcomes, and limitations of the virtual patients were analyzed. Twenty-seven full texts were reviewed, including 17 case reports, 10 non-randomized controlled experimental studies, 75 patients, and 3 phantoms. Few studies showed a low risk of bias. Dynamic data included real-time jaw motion, simulated jaw position, and dynamic facial information. Three to five types of information were integrated to create virtual patients based on diverse superimposition methods. Thirteen studies showed acceptable dynamic techniques/models/registration accuracy, whereas 14 studies only introduced the feasibility. The superimposition of stomatognathic data from different information collection devices is feasible for creating dynamic virtual patients. Further studies should focus on analyzing the accuracy of four-dimensional virtual patients and developing a comprehensive system.

## 1. Introduction

Digital workflows are becoming more accurate in dental medicine because of technological innovations. Transferring intraoral and extraoral data to a virtual environment is the first step in digital treatment. Currently, digital information can be captured in different ways, including desktop scanners (DS), intraoral scanners (IOS), facial scanners (FS), cone beam computed tomography (CBCT), computed tomography (CT), cephalometry, and photography. These methods can produce different file formats, such as standard tessellation language (STL), object code (OBJ), polygon (PLY), and digital imaging and communications in medicine (DICOM). A three-dimensional (3D) virtual patient can be created after the alignment and fusion of various data formats, including information about a real patient’s teeth, soft tissues, and bones [1]. Thus, if real patients are indisposed, dental treatment plans could still be realized in virtual patients reducing chair time and patients’ appointments.

Currently, investigators focus on static virtual patients, with improved gains through new materials, automation, and quality control [2,3]. However, static simulated patients cannot reflect real-time changes. The stomatognathic system comprises the skull, maxilla, mandible, temporomandibular joint (TMJ), teeth, and muscles, and changes in one part will cause synergistic changes in the others [4]. Therefore, integrating TMJ and mandible movement, occlusal dynamic, and soft tissue dynamic information (such as muscle movement and facial expression) to construct four-dimensional (4D) virtual patients is required in the future [5]. Four dimensions use time to express action; thus, four-dimensional patients with temporal information help understand the dynamic interactions of anatomical components under functional activities such as chewing, speech, and swallowing.

The first step in creating a 4D virtual patient is digitalizing the motion data. The virtual facebow (VF) and virtual articulator (VA) combination can facilitate positional relationship replication between the skull and jaws, simulating mandible movements [6]. In addition, a jaw motion analyzer (JMA) moves the digitized dentitions along paths in the computer, helping to visualize kinematic occlusion collisions and the condyle trajectory [7,8]. Information such as the smile line, lip movements, and facial expressions is essential to ensure functional outcomes and aesthetic performance and to construct a pleasant smile [9]. Finally, in various computer-assisted design and computer-aided manufacturing (CAD/CAM) programs, 4D simulated patients with complex movements are built. Currently, 4D virtual patient types vary according to clinical needs. Few studies have comprehensively analyzed the existing 4D dental virtual model construction techniques.

Therefore, this systematic review aims to summarize the current scientific knowledge in the dental dynamic virtual patient field to guide subsequent related research.

## 2. Materials and Methods

### 2.1. Eligibility Criteria

This systematic review followed the Preferred Reporting Items for Systematic Reviews and Meta-Analyses (PRISMA) guidelines [10,11,12]. (Appendix A) The present review was not registered because it belongs to the literature reviews that use a systematic search, which PROSPERO does not accept. The focus was on the technique, accuracy, and application of dynamic multi-modal data fusion to create four-dimensional virtual patients in dentistry. The criteria for study selection were (1) the possibility of creating a 4D dental dynamic virtual patient analyzing at least one patient or phantom; (2) possible integration of two or more digital methods, at least one of which captures dynamic information; (3) availability of the methods and devices used; (4) review articles, opinion articles, interviews, charts, and non-English articles were excluded from this systematic review. The PICOS terms were defined as population (P), four-dimensional virtual patient; intervention (I), dynamic digitization technology; comparison © was omitted because the current review was not expected to have randomized controlled trials or relevant controlled trials; outcome (O): dental applications or accuracy analysis; setting (S): multi-modal data fusion.

### 2.2. Information Sources

The literature search was conducted by reviewing three online databases for eligible studies: PubMed, Medline (Web of Science), and Cochrane Library. The references of the full-text articles were additionally screened manually for other relevant studies. A four round “snowball procedure” was carried out to identify other published articles that met the review’s eligibility criteria. (Appendix A) The “snowball procedure” is a multi-round forward screening, after the full-text screening, to search the eligible papers from the reference lists of the included papers. Once a new study/reference is included, its references are called snowball papers, which will undergo a new round of snowball screening. This procedure ends only when no snowball papers can be included in the last round [13].

### 2.3. Search Strategy

The first search in the database was performed on 5 August 2022. The search strategy was assembled using Medical Subject Headings (MeSH) terms and free-text words. Search terms were grouped according to the PICOS principle (Table 1). Weekly literature tracking was then conducted separately in the three databases using the above search terms to obtain the latest relevant literature.

### 2.4. Study Records

After the first duplicate check in NoteExpress, the articles were imported into the Rayyan website [14] for the second duplicate check. Titles and abstracts were screened independently by two reviewers (YY and QL) on Rayyan [15], a tool to filter titles and abstracts effectively and to collaborate on the same review. Disagreements were resolved by discussion. For controversial articles labeled as maybe, reviewers discussed including them after reading the full text. After screening, full-text articles of selected titles and abstracts were acquired and read intensively by two reviewers to determine eligible articles. All authors discussed the remaining controversial articles to obtain a consensus. Snowball articles were included from the full-text references and were selected following the same principle.

### 2.5. Data Extraction

The following parameters were extracted from the selected full-text articles after the selection process: Author(s) and year of publication, study design, sample size, methods (including file format), manufacturer software programs, information, type of superimposition, scope, outcomes, and limitations.

### 2.6. Evaluation of Quality

The Joanna Briggs Institute (JBI) Critical Appraisal Checklist for quasi-experimental studies [16] and case reports [17] was used for the non-randomized controlled experimental studies/case reports. Two reviewers (YY and QL) independently assessed the methodological quality of the included studies. For every question in the checklist, except Q3 in the Checklist for Quasi-Experimental Studies, a yes answer means the question is low risk. Studies that met 80–100%/60–79%/40–59%/0–39% of the criteria were considered to have a low/moderate/substantial/high risk of bias, respectively [13]. In cases of disagreement, the decision was made by discussion among all authors.

## 3. Results

### 3.1. Search

The systematic search was completed on 30 November 2022. The QUORUM diagram details are shown in Figure 1. The snowball procedure is illustrated in Appendix A. The search yielded 1776 titles, 173 titles and abstracts were identified. Subsequently, 19 full texts were selected by two reviewers, of which 550 references underwent four rounds of citation checks using the snowball procedure, yielding 14 articles. Six controversial articles were excluded from the 19 full-text articles based on the outcomes of the discussion between all authors. The reasons for exclusion were as follows: single dynamic data (*n* = 1), non-dynamic patients (*n* = 4), and no virtual patient built (*n* = 1). Finally, 27 articles were included in the systematic review; the reasons for excluding other papers are shown in Figure 1.

### 3.2. Description of Studies

The characteristics of the included studies are presented in Table 2. The included publications were dated from 2007 to 2022, without intentional time restriction, since the 4D virtual patient is a new technology recently proposed. The review included 17 case reports and 10 nonrandomized controlled experimental studies. No randomized controlled trials (RCT) were found. Most of the 27 studies had only one subject/phantom, except for 2 [18,19]. In total, 75 patients and three phantoms were included in creating 4D virtual patient models.

Two or more methods, including static information and dynamic information collection devices, can acquire different formats of 3D data. The present article focuses on dynamic data (Table 3); 10 studies acquired real-time jaw-motion data, 4 analyzed the dynamic facial information, 13 simulated the jaw position, and 2 examined the coordinated movement of the masticatory system. Three to five types of information acquired from the above data were integrated to create 4D virtual patients. Additionally, the included studies focused on different clinical scopes: prosthetic dentistry (*n* = 19) (including implant dentistry (*n* = 4)), maxillofacial surgery *(n* = 8), and orthodontics (*n* = 5).

### 3.3. Risk of Bias in Included Studies

Appendix A describes the risk of bias assessment in the 10 non-randomized controlled experimental studies. All had a low risk of bias for Q1, Q4, Q7, and Q8 of the JBI Critical Appraisal Checklist. For Q2, Q3, and Q9, 80%, 90%, and 60% of the studies had a low risk of bias, respectively. For the overall risk, 90% of the studies showed moderate risk and 10% showed substantial risk (Figure 2a). Appendix A describes the risk of bias assessment of the 17 case reports. All these had a low risk of bias for Q5, Q6, Q7, and Q9 of the JBI Critical Appraisal Checklist. Except for four studies that indicated low risk, Q4 was not applicable in most studies where specific diseases were not crucial during the construction of the simulated model. However, for Q1, Q2, and Q3, 76.5%, 69.5%, and 52.9% of the case reports, respectively, showed a high risk of bias. For the overall risk, 52.9%, 29.4%, 11.8%, and 5.9% showed high, moderate, substantial, and low risk, respectively (Figure 2b).

### 3.4. Outcomes

#### 3.4.1. Dynamic Data Collection Methods of 4D Virtual Patients

1.Mandibular movement (jaw motion)

Ten studies acquired real-time jaw-motion data. Only two mentioned the dynamic file format (Moving Picture Experts Group 4 (MP4) [20]/text file (TXT) [20,38]). One study captured videos of mandibular movements using a target-tracking camera [20]. Two studies introduced FS and targets to record jaw movements and the accumulated movement paths [21,31]. The kinematics of the occlusion/condyle can be simulated by combining jaw tracking with DS/CBCT. The accuracy of the FS + targets method, evaluated by the distance between the targets, showed a value of 4.1–6.9 mm (a minor error compared with that of laboratory scanners [78 mm]). Three studies used an electromagnetic (EM) system to track the positions of the maxillary bone segment (MBS) and the reference in physical space [25] or the mandibular proximal segment (MPS) and condyle position [26], or mandibular repositioning and occlusal correction [18]. One study used an optical analyzer to track the position of light-emitting diodes mounted on facebow and jaw movements [4]. Three studies acquired mandibular movements [29,35,38], condylar motions [29,38], and collision detection [38] by the ultrasonic system. The ultrasound Arcus Digma system has an accuracy of 0.1 mm and 1.5°, and its pulse running time was converted to 3D coordinate values and saved as TXT files [38].

2.Dynamic faces

One study [23] used a smartphone facial tracking app to track lip dynamics. Three studies acquired semi-dynamic faces using FS and image processing software. Semi-dynamic facial information refers to dynamic bits of facial details [42] or faces at different times [34,43] instead of real-time facial changes.

3.Positional relationships of the dentition, jaws, skull, and TMJ (jaw position)

Thirteen studies obtained positional relationships to help simulate jaw movements on the VA. First, the facial reference system helped to locate the natural head position (NHP), a reliable plane to align the skull and VA hinge axis [22,30]. The common VF techniques involve transferring the maxillary dentition to the FS, guided by an Intraoral transfer element (IOTE) (i.e., facebow fork) or a scan body, and to the FS without an IOTE/scan body [19,22,24,32,33,37,40,42]. A modified IOTE combined with LEGO blocks, trays, and impressions enabled a convenient transfer procedure [32,33]. The VF showed average trueness of 1.14 mm and precision of 1.09 mm based on the FS of a smartphone [19]. There is a difference between the virtual transferred maxillary position and its real position (a trueness of 0.138 mm/0.416 mm and a precision of 0.022 mm/0.095 mm were obtained using the structured white light (SWL)/structure-from-motion (SFM) scanning method). This difference is mainly caused by the registration error, which may be reduced by different alignment methods and IOTE [37]. Special VF transfers the relationship by anatomical points/marker planes, reducing the errors by omitting the traditional facebow transfer [36]. The FS can be replaced by photographs [30,41]/CBCT [27,28]; however, the accuracy was not calculated. Once VF transfer is completed, VA systems are available for jaw motion simulation.

4.Coordinated movement of the masticatory system

Two studies used finite element (FE) software to analyze mandibular movements, masticatory muscle performance, occlusal force [38], and stress distribution of TMJ [39].

#### 3.4.2. 4D Superimposition Techniques of Virtual Patients

The main components of stomatognathic information include static information such as the skeletal components (SK) of the skull, jaws, and TMJ, dentition (DENT), dynamic information such as mandibular movement (MM), jaw position (JP), occlusal analyses (OA), and motion of TMJ soft tissues and muscles. Moreover, soft tissues of the face (SF) can be either in a dynamic or static form. Three to five of these components were combined to create virtual models based on 4D superimposition techniques. These models are created using various software systems.

1.Five types of information superimpositions

SK + DENT + MM + SF + TMJ soft tissues/SK + DENT + MM + OA + muscles

Two studies created models [29,38] including the skeletal components from CT (DICOM format), dentitions from DS (STL format), and jaw movement from the Jaw motion tracker (JMT) (TXT format [38]). Lszewski et al. [29] added MRI’s facial and TMJ soft tissues to the above data. The virtual model’s superimposition and real scenes were based on an algorithm and fiducial markers. The module’s accuracy requires further validation before clinical application. Dai et al. [38] included masticatory muscles and occlusal analyses in their model. The regional registration method was used to superimpose the dentition in the occlusion. Their FE model showed an accuracy (≤0.5 mm) similar to that of the T-Scan.

2.Four types of information superimpositions

SF + DENT + OA + MM/JP:

Two studies integrated facial tissues from FS, dentition from DS [21]/IOS [19], occlusal contact analyzed by the CAD software, and mandibular movement tracking from FS + targets [21]/jaw position from VF + VA [19]. The alignment of the maxillary cast, scan body, and facial tissues was based on the points of the teeth [21]. The superimposition of FS and CBCT was based on the facial surface. The deviation of their models was 1 mm in linear distance and 1° in angulation [19].

SK + SF + DENT + JP:Six studies [24,27,28,32,33,34] constructed virtual patients mounted on the VA with faces from the FS (OBJ [24]/STL [27]/PLY [28] format)/photographs [34], bones from CBCT/CT (DICOM format), and dentition/prosthesis from the IOS (STL format). The models of three studies [24,27,28] were centric relation occlusion (CRO) and vertical dimension of occlusion (VDO). The prosthetic outcomes demonstrated a good fit, occlusion, and esthetics. Three studies created models based on points and fiducial markers [24,32,33]. One study [32] aligned the models with CBCT using the Iterative Closest Point (ICP) algorithm. The errors in tooth registration were less than 1 mm, whereas those of the nasion, alares, and tragions were 0.83 mm, 0.77 mm, and 1.70 mm, respectively. Further research is required to reduce this discrepancy and distortion. Granata et al. [42] created a virtual patient with faces in a smiling, open mouth, and maximum intercuspation (MICP) positions from FS (OBJ format), dental arches from IOS (PLY/STL format), and bones and jaw position from CBCT with the occlusal registration device Digitalbite (DGB). The superimposition was based on fiducial markers and a best-fitting algorithm.SK + SF + DENT + MM:

Kwon et al. [31] introduced a virtual patient with skeletons from CBCT, dentitions from DS, and a face combined with mandibular motions using FS + targets. The alignment was based on triangulated mesh points and a transformation matrix. The tracking system stability and reproduction were acceptable compared with routine VF transfer. Noguchi et al. [43] created virtual models integrating the mandible, TMJs, and outline of the soft tissue from cephalometry and dentition from DS and traced their movement. The projection-matching technique is based on the contour line of the projection image, and the registration error is the same as that in conventional cephalometry.

SK + DENT + MM + TMJ soft tissues:

Savoldelli et al. [39] combined the bone components of jaws and TMJs, dental arches from multi-slice CT, and soft tissues of TMJs when the jaw was opened 10 mm. Then, the joint discs’ boundary conditions and stress distribution were analyzed showing a high level of accuracy (stress levels of the model [5.1 MPa] were within the range of reported stress [0.85–9.9 Mpa]).

3.Three types of information superimpositions

SK + DENT + MM:

Zambrana et al. [20] constructed a virtual patient with jaws and TMJs from CBCT (DICOM format), dentition with maxillomandibular relationship from IOS (STL format), and mandibular movements from a target tracking video (MP4 format). Registration was based on the surface and points. Five studies combined bones from CT [4]/MDCT [25,26]/CBCT [35]/cephalograms [18], dentition from DS [4,25,26]/CBCT [35], and positions of MBS [25]/MPS [26]/mandibular movement [4,18,35] from the JMT. Integrations [4,18,25,26,35] were based on fiducial markers, and some were combined with the least-squares method [4]/ICP algorithm [25,26]. The technique of Fushima et al. [18] showed high accuracy (the minor standard deviations (SD) <0.1 mm). The two methods showed no significant difference between the actual and measured positions [25]. The condylar landmark results showed high accuracy (differences between MPS models and those between CT models were 1.71 ± 0.63 mm and 1.89 ± 0.22 mm, respectively) [26]. He et al. [35] analyzed the condyle position, which was more accurate than CBCT, with records showing high precision over three days.

SF/SK + DENT + JP:

One study [36] combined bones from CBCT with mandibular position from VA and VF and dentition in CRO/MICP from IOS to simulate a patient. Six studies [22,30,37,40,41] described methods to create models with faces from FS [22,23,37,40]/2D photograph [30,41], dentition models from IOS [22,23,30,41]/DS [37], and mandibular position using VF and VA. Kois et al. [30] used the “Align Mesh” tool to align data. The Alignment of the casts, scan body, facial scan, or alignment of the facebow forks was based on points [22,41]. IOS and CBCT/FS were registered based on points and the ICP algorithm [36,37] or fiducial markers of the scan body [23,40].

4.Software programs to create the virtual patients

An open-source program (Blender 3D; Blender Foundation, Amsterdam, Netherlands) can import MP4 dynamic information directly. The MP4 file can be transferred into TXT format using a direct linear transform (DLT) algorithm, which facilitates the integration [20], marking the reference points to align the models acquired from other CAD software programs using Python [19]. In this program, STL/PLY files can be transferred to the OBJ format to facilitate the fusion of different formats [23].

Exocad (Exocad; exocad GmbH, Darmstadt, Germany) is the most common multifunction CAD software used to build virtual patients. It can align the IOS/DS with the FS, with/without CBCT, guided by the scan body [22,40]/IOTE [19]/gothic arch tracer, wax [27,40]/DGB [42], positions NHP, transfers facebow, and finally integrates the model into the VA. Exocad can also analyze the occlusal discrepancy between MIP and CR [21,24] or design a tooth-supported template [28]. Finally, restorations can be created based on the VDO and the occlusal plane [34]. Other software programs for constructing the 4D virtual patients are presented in Table 4.

FE modeling and analysis software can convert 4D virtual models created by other software into numerical models and analyze dynamic/static components. The FE software commonly used for 4D virtual models is ANSYS (ANSYS Inc., Canonsburg, PA, USA) [38]/FORGE (Transvalor, Glpre 2005, Antibes, France) [39]. Processing software such as AMIRA (Visage Imaging, Inc., SD, CA, USA) are often used to obtain surface and volume meshes. The volumes of the anatomical components were input to FE software and meshed as the element; subsequently, the mesh quality and nodes’ quality were verified. The accuracy of a FE model is determined using geometric models.

#### 3.4.3. Clinical Applications of 4D Virtual Patients

The 4D virtual patient is built to apply to different clinical scopes.

1.Application in prosthetic dentistry and dental implant surgery

In this field, the 4D virtual patient mainly involves locating the jaw positions and condylar axes, obtaining functional data, simulating the mandible and condyle trajectories, and analyzing occlusion. Traditional restoration processes focusing only on static occlusion may lead to poor occlusal function and TMJs disorder. However, for digital workflows, VF techniques locate the position of jaws and condyles [19,22,23,24,27,28,30,32,33,37,40,41,42], and VA [22,24,27,28,30,36,40] or JMA [20,31,35] helps simulate or record patient-specific mandibular and TMJ kinematics. These procedures obtain the correct MICP, CRO, and VDO for coordinated dental implants and restorations in a stable position [24,27,28,36]. The dynamic occlusal analysis allows the detection of occlusal interference during eccentric movements to design anatomic prostheses [4,19,21]. Additionally, the condylar motion trajectory and mandibular movement pathway can help diagnose and treat TMJ diseases and facilitate occlusal reconstruction [20,31]. Furthermore, adding facial information to the virtual patient is beneficial to harmonizing the prosthesis with the face, ensuring an aesthetic effect [23,30,34,41,42]. CAD systems perform the above digital analysis and design, and finally, dock CAM for guide template and restoration fabrication. The innovative workflows of 4D virtual models are particularly suitable for complex implant rehabilitation [24,27,28] and complete denture restoration [40], resulting in excellent repair results with patient satisfaction.

2.Application in maxillofacial surgery

Traditional orthognathic surgery planning is effective but time-consuming, and many factors, such as the occlusal recording and mounting, affect the accuracy. Computer-assisted orthognathic surgery is an interdisciplinary subject that combines signal engineering, medical imaging, and orthognathic surgery to improve efficiency. In addition to facial esthetics, reducing mandibular spin is essential to obtaining stable skeletal and occlusal outcomes and preventing temporomandibular disorders. Therefore, studies have improved accuracy and minimized spin using 4D virtual models in surgical systems.

This article mainly included planning [4,18,29,35,36,43] and navigation systems [25,26]. The ACRO system integrates modules for planning, assisting surgery, and bringing information from virtual planning to the operating room [29]. The Aurora system (Aurora, Northern Digital Inc., Waterloo, ON, Canada) uses augmented reality (AR) to locate bone segments and condylar positions [25,26]. The 4D analysis system TRI-MET (Tokyo-Shizaisha, Tokyo, Japan) was used to simulate mandibular motion, condylar to articular fossa distance, and occlusal contact [4]. The mandibular motion tracking system (ManMoS), a communication tool for operational trial and error, predicts changes in occlusion and repeatedly determines mandibular position [18]. The SICAT system (SICAT, Bonn, Germany) can show the motion of the incisors and condyles during mandibular movement, thereby avoiding additional radiation exposure [35]. The simulation system based on cephalometry allows the location of 3D bone changes without CBCT, reducing radiation and errors in manual pointing [43].

3.Application in orthodontics

Most of the included papers have cross-disciplinary applications. The VF, VA, and JMA to locate the mandible and condyles are also crucial for orthodontics. This helps reconstruct stable and balanced occlusion in the optimum position and prevents recurrence and TMJ symptoms. The simulated position of the incisors, jaws, and soft tissue provides a visual treatment objective. As mentioned above, analysis of pre- and post-operative tissue changes [34], orthodontic-orthognathic planning [4,43], and dynamic facial information of virtual patients are also applicable to aesthetic orthodontics plans and outcomes.

Traditional orthodontic treatment often focuses on dentition and bone problems in three dimensions but ignores the improvement of mastication efficiency and TMJ health. Quantifying masticatory function is essential for occlusal evaluation and orthodontic tooth movement. The directly mentioned application of the dynamic patient models in orthodontics mainly involves the analysis of masticatory muscles, stress distribution in the articular disc, mandibular movements, and occlusion [38,39]. FE methods can simulate dynamic masticatory models, including muscle forces to the teeth, to determine the magnitude and direction of the bite force. High-resolution FE models analyzed the stress distribution and symmetry of TMJ and the boundary conditions of mandibular during the closure process. Further studies are expected to enable the prediction of various stress loads on the TMJ disc in the context of mandibular trauma, surgery, or dysfunction.

## 4. Discussion

With the development of computer-aided design/computer-aided manufacturing (CAD/CAM) technology, virtual patient construction, incorporating multi-modal data, has been widely used in multiple fields of dentistry. Building a 4D virtual dental patient using dynamic information is of great interest. The present review revealed the methods, manufacturer software programs, information, registration techniques, scopes, accuracy, and limitations of existing approaches to dynamic virtual patient construction. Data from multiple sources and formats were captured using various methods and programs. Specific alignment methods integrate various types of information to build 4D virtual patients in different clinical settings.

### 4.1. Dynamic Data Collection Methods

The JMT system, FS + targets, and target tracking camera were used to acquire real-time jaw motion data. The EM JMT system uses a magnetic sensor to track jaw motion, bone segment, and condyle positions and a receiver to detect movement, which is popular in minimally invasive surgeries [18,25,26]. Nevertheless, electromagnetic interference can affect the device’s accuracy [44]. An optical JMT system can display condyles and mandibular movement trajectories. The limitation of this method is the strict conditions and motion restriction by a sizeable facebow [4,20]. Ultrasonic JMT systems transfer acoustic signals from the transmitter into spatial information to record movements, which may be vulnerable to environmental conditions [29,35,38]. FS or photographs combined with targets can simultaneously capture jaw motion and the face and show a small error [21,31,41]. The mobile phone’s camera, connected to a marker board, captures movements inexpensively and conveniently but without a test of the accuracy [20]. These methods capture much information, such as mandibular movements and kinematics of the condyles, in all degrees of freedom (including excursive movements, maximum mouth opening, protrusion, and lateral excursions). However, few studies have reported real-time jaw motion accuracy and file format.

Integrating kinematic digital VF with VA makes capturing jaw movements with acceptable accuracy possible [19,37]. Most VA assembly procedures include a digital impression of dentition, occlusal recording, VF transfer of the maxilla position to the skull, and mounting the models to VA. Various software [19,22,24,27,28,30,33,34,36,37,40,41,42] include the VA procedure. Although this approach does not present motion in real-time, it is compatible with file formats and requires more information. The FE method accurately simulates the masticatory system [38,39]. In addition, studies on the fusion of dynamic facial information in 4D virtual patients are lacking.

### 4.2. 4D Superimposition Techniques

The 4D superimposition technique integrates three to five types of information to create dynamic patients. The technology’s superimposition methods, software, and outcome varied among different research.

#### 4.2.1. Superimposition Methods

Image fusion and virtual patient creation are based on selecting the corresponding marker for the superimposition of data from multiple sources. The construction of a simulated model may involve multiple alignment processes. The alignment methods in the included articles were mainly based on points [20,21,22,24,32,33,36,37,40,41], fiducial markers [4,18,22,23,24,25,26,27,29,31,32,33,35,40,41,42], surface [19,20], and anatomical structure [39,43]. Registration based on additional attached fiducial markers is also a point-based registration method. Some algorithms, such as the ICP [21,26,27,32,33,36,37], best-fit [19,22,29,41,42], and least squares methods, were used to help the registration processes [4]. The specific alignment techniques used in each study are presented in Table 2.

Real-time mandibular movement data integration was mainly based on the fiducial marker [4,18,26,27,29,35], but the specific integration principles were not described. The alignment step affected the final virtual model’s accuracy. However, only two studies have evaluated the accuracy of alignment methods [32,34]. Therefore, further studies should introduce registration methods for the dynamic data of 4D patients and quantify the accuracy and optimization methods for every alignment step.

#### 4.2.2. Software Programs

The 4D virtual model ensures that a comprehensive model contains the required data, while the FE analysis software chunks the complex model into simple units connected by nodes, facilitating simple algorithms for the analysis and interpretation of complex data. Although few FE software programs analyzed 4D models, several FE-related programs were used to analyze 3D models. For example, Hypermesh (Altair, Troy, MI, USA) is an important preprocessing software, and Abaqus (Abaqus Inc., Providence, RI, USA) is a common FE analysis software that interprets geometric models [45,46].

Currently, commercial software is available for dynamic dental virtual models; however, the principle is not specified, and the accuracy needs further improvement. Various computer software packages have made it convenient to use diverse clinical information. However, creating patient models by superimposing multi-modal data is still new. Integrating data from diverse file formats may be incompatible and inaccurate. Other limitations are tedious processes, requiring different hardware and software to acquire and analyze data, and expensive fees.

#### 4.2.3. Outcome of the Technology

Most recent studies on 4D patients have included only one patient or phantom for feasibility exploration. Further studies are required to increase the sample size. In that regard, fourteen studies did not evaluate their results’ reliability [4,20,21,22,23,24,27,28,30,34,36,40,41,42], six assessed different dynamic techniques’ accuracy [19,29,31,33,35,37], five evaluated the simulated model’s accuracy [18,25,26,38,39], and two reported the registration accuracy [32,43]. Owing to the differences in the sources, formats, integration methods, and use of different 4D virtual patients, there is still a need for a unified standard to assess 4D virtual patients’ accuracy.

### 4.3. Clinical Applications

Depending on the clinical needs, 4D virtual patients integrate static virtual patients with dynamic information. The present review focuses on applying dynamic information of the virtual models: the establishment of occlusion in a stable mandibular and condylar position is of great benefit for restoration design, implant planning, orthodontic treatment, and orthognathic surgery. In addition to analysis of masticatory function and occlusal interference in the functional state, real-time jaw and joint movements are also useful for cause analysis and treatment of TMJ disorders and intraoperative navigation in implantology and orthognathic. The dynamic face of the virtual models facilitates the smile design. Overall, dynamic virtual patient models facilitate pre-treatment planning, intraoperative assessment, and stable, healthy, and aesthetic treatment outcomes for actual patients. Virtual patient construction achieves an intuitive presentation, facilitating communication and clinical decision-making in dentistry.

There are some limitations to this systematic review. First, the included studies were either non-randomized experimental studies or case reports. Few studies showed low overall risk. Therefore, the scientific level of clinical evidence is lacking. Appropriate statistical methods are needed to evaluate non-randomized experimental studies. The case reports lacked demographic characteristics, medical histories, and current clinical conditions. Second, the total sample size was 78, which needs further expansion. Third, there was considerable variation in subjects, interventions, outcomes, study design, and statistical methods across the included studies. Due to high heterogeneity, we did not conduct a meta-analysis but only qualitatively discussed the technique and application. Fourth, a few included articles were a series of related studies done by the same authors, which may increase the bias of the results. Although all criteria are met, such cases should be avoided in future studies. Finally, the number of included snowball papers was more than that obtained by electronic searches. There were no suitable articles in the weekly literature tracking. Thus, our search strategy needs to be improved.

High-quality clinical studies such as RCT should be conducted in 4D virtual patients to ensure an appropriate design, sufficient sample size, and less heterogeneity. Registration methods for dynamic data should be introduced and optimized. Future investigators should evaluate the accuracy of the currently available techniques to create 4D dynamic virtual patients. Furthermore, it is better to establish a unified evaluation standard conducive to quantitative analysis. In addition, integrating all of the required information within one system should also be considered.

## 5. Conclusions

Based on the included articles, the following conclusions were drawn:Dynamic data collection methods of 4D virtual patients include the JMT, FS + targets, and target tracking camera to acquire real-time jaw motion, VF and VA to simulate jaw position, facial tracking systems, and FE programs to analyze the coordinated movement of the masticatory system.Superimposition of the skeleton, TMJs, soft tissue, dentition, mandibular movement/position, and occlusion from different static/dynamic information collection devices in various file formats is feasible for 4D dental patients.Four-dimensional virtual patient models facilitate pre-treatment planning, intraoperative assessment, and stable, healthy, and aesthetic treatment outcomes in different clinical scopes of dentistry.There is a lack of well-designed and less heterogeneous studies in the field of 4D virtual patients.Further studies should focus on evaluating the accuracy of the existing software, techniques, and final models of dental dynamic virtual patients and developing a comprehensive system that combines all necessary data.

## Figures and Tables

**Figure 1 jfb-14-00033-f001:**
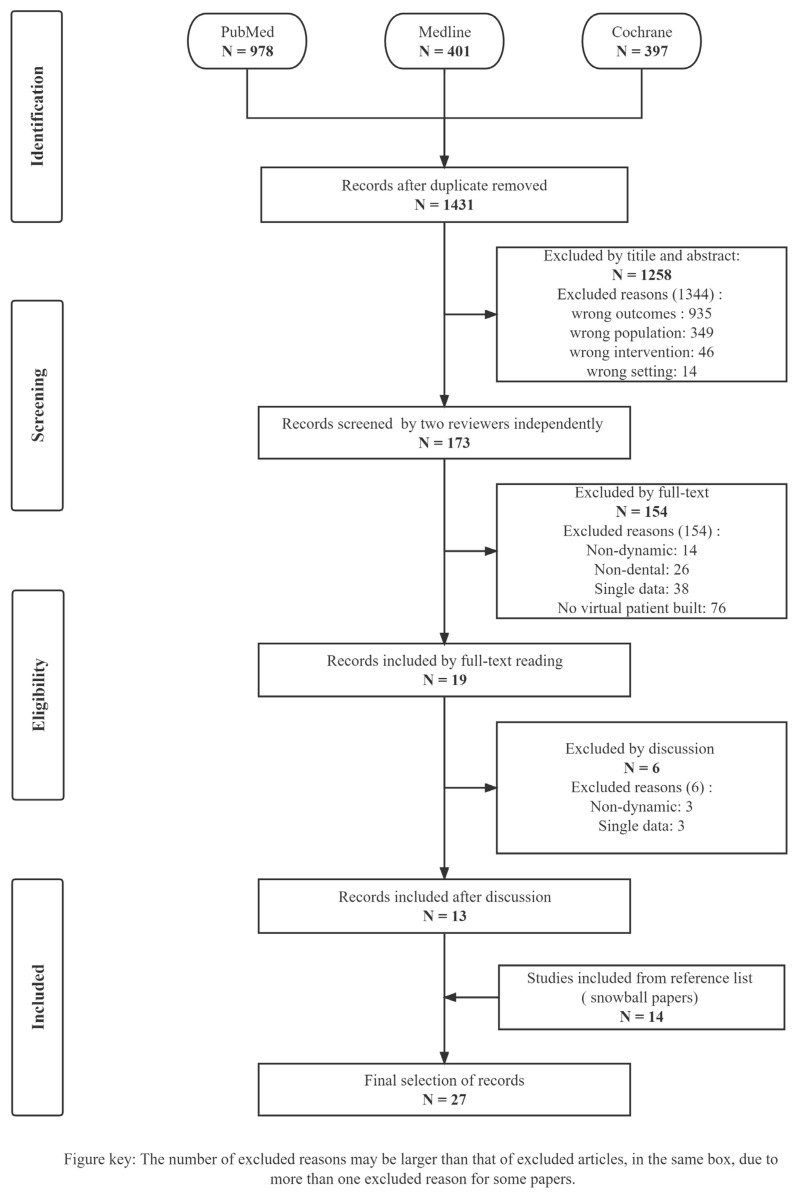
PRISMA flow chart. This diagram describes the identification, screening, exclusion reasons, and included procedures of 27 included articles. The number of excluded reasons exceeds that of excluded articles only at the screening phase, where the total number of reasons excluded was 1344, but 1258 articles were excluded.

**Figure 2 jfb-14-00033-f002:**
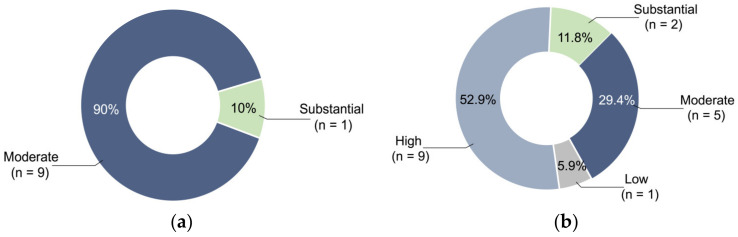
Percentage of different risk levels: (**a**) For the overall risk in non-randomized controlled experimental studies, 90% showed moderate risk and 10% showed substantial risk. (**b**) For the overall risk in case reports, 52.9%, 29.4%, 11.8%, and 5.9% showed high, moderate, substantial, and low risk, respectively.

**Table 1 jfb-14-00033-t001:** Overview of Electronic Search Strategy.

Subjects	Answers
Database	PubMed, Medline (Web of Science), and Cochrane library
#1 Population	“Patient simulation” [MeSH] or “computer simulation” [MeSH] or “Patient Simulations” or “computer simulations” or “computerized model *” or “Computer Model *” or “Virtual patient *” or “digital patient *” or “simulation patient *” or ((“4D” OR “4-D” or “4 dimension *” or “4-dimension *” or “four D” or “four-D” or “four dimension *” or “four-dimension *”) and “patient *”)
#2 Intervention	“Dental Articulators” [MeSH] or “dynamic” or “Dental Articulator” or “Articulator, Dental” or “Articulator *” or “Condylar movement” or “ condylar position” or “computer aided diagnosis axiograph” or “Diagnosis, Computer Assisted” or “Computer-Assisted Diagnosis” or “Computer Assisted Diagnosis” or “Computer-Assisted Diagnosis” or “Diagnoses, Computer-Assisted” or “Jaw motion” or “jaw movement” or “mandibular movement” or “Face bow” or “facebow” or “Electromyography” [MeSH] or “Electromyographies” or “Surface Electromyograph *” or “Electromyography, Surface” or “Electromyogram *” or “EMG” or “EMCP” or “photogrammetry” [MeSH] or “Stereophotogrammetr *” or “Radiostereometric Analysis” or “ facial scanning” or “Mastication” [MeSH] or “chewing” or “Dental Occlusion” [MeSH] or “Occlusion?, Dental” or “Dental Occlusions” or “Occlusal Plane *” or “Plane?, Occlusal” or “Canine Guidance” or “Guidance, Canine” or “Occlusal Guidance *” or “Guidance, Occlusal.”
#3 Outcome	“Dimensional Measurement Accuracy” [MeSH] or “Dimensional Measurement Accurac *” or “Measurement Accuracy, Dimensional” or “accuracy” or “precision” or “reliability” or “Validity” or “stomatology” or “tooth” or “oral” or “dental” or “Dentistry” or “orthodontics *” or “prosthodontics *” or “implant dentistry*” or “orthognathic *” or “Maxillodental *” or “orthognathic* ” or “maxillofacial surgery *” or “plastic surgery *”
#4 Setting	“multi-modal” or “multi-mode” or “multi-modality” or “multimodal” or “multiple-modal” or “multiple-mode” or “multiple-modality” or “multi-source” or “multisource” or “fusion” or “integration” or “superimposition” or “merging” or “registration” or “alignment” or “calibration”
#1 and #2 and #3 and #4	

**Table 2 jfb-14-00033-t002:** Information on the 27 included studies to compose dynamic simulated dental patient.

Author Ref./Year	Study Designs (Humans/Phantom)	Sample Size (n)	Methods (+File Format)	Manufacturer Software Programs	Information	Types of Registration/Superimposition	Scopes	Outcomes	Limitations
Zambrana N [20]/2022	Case report: method description (Humans)	Unknown	1. CBCT (DICOM) 2. IOS(STL) 3. Target tracking camera (MP4) 4. DS 5. Open-source CAD software	1. (Promax 3D Mid, Planmeca OY, Helsinki, Finland) 2. (TRIOS3; 3Shape A/S, Copenhagen, Denmark) 3. Camera of a mobile phone (iPhone 7; Apple Inc., Cupertino, CA, USA) 4. (Swing; DOF Inc., Seoul, Republic of Korea) 5. (Blender 3D; Blender Foundation, Amsterdam, Netherlands)	1. TMJs, maxilla, and mandible 2. Dentitions and the maxillomandibular relationship 3. Mandibular movements 4. Maxillomandibular relationship, including the marker boards 5. 4D virtual patient with mandibular kinematic path and TMJ kinematic path	Markerboards + marker board from occlusal registration (IOS + DS): point-based	Prosthetic Dentistry	Feasibility	1. Lack of a rigorous validation 2. Lack of comparison with other jaw-tracking systems and software programs
Kim JE [21]/2019	Case report: method description (Humans)	Unknown	1. DS 2. FS + targets 3. Image registration software 4. CAD software	1. (Identica hybrid, Medit Inc., Seoul, Republic of Korea) 2. (Rexcan CS2, Medit Inc., Seoul, Republic of Korea) 3. (EzScan8, Medit Inc., Seoul, Republic of Korea) 4. (Exocad; exocad GmbH, Darmstadt, Germany)	1. Virtual casts 2. Anterior part of dentition, face, and target position 3. Four types of antagonist mesh + models superimposing dental cast data on facial scan data 4. Occlusal contacts in the MICP and occlusal interference during eccentric movement	FS + DS: point-based(teeth) + horn alignment algorithm + ICP algorithm	Prosthetic dentistry	Feasibility	1. Time-consuming 2. The need to maintain the retractor 3. Lack of validation of accuracy
Revilla-Leon M [22]/2022	Case report: method description (Humans)	Unknown	1. FS(PLY) 2. IOS(STL) 3. VA 4. DS(PLY) + scan body system 5. CAD software	1. (InstaRisa Facial Scanner; InstaRisa, Clovis, CA, USA) 2. (TRIOS4, wireless, v21.2.0; 3Shape A/S, Copenhagen, Denmark) 3. (Panadent Articulator; Panadent, Colton, CA, USA) 4. (Medit T500; Medit Inc., Seoul, Republic of Korea) + (Kois Scan Body System; Kois Center, LLC, Seattle, WA, USA) 5. (DentalCAD 3.0, Galway; exocad GmbH, Darmstadt, Germany)	1. Static and smiling face 2. The dentitions and a maxillomandibular registration 3. Jaw-correction function 4. NHP + registration tool 5. Virtual patient with a digitized scan body, maxillary digital scan, and reference facial scan	1. Maxillary cast + the scan body (DS + IOS): point-based + best-fit alignment (teeth as common information) 2.Scan body + reference face (DS + FS): point-based (scan-body as common information) 3. Reference face + the smiling face (FS + FS): point-based (facial markers as common information)	Prosthetic dentistry	Feasibility	1. Lack of validation of accuracy 2. Factors affecting accuracy: 1). the eye closure due to the intense light 2) color and surface texture of the scan body system
Li J [19]/2022	Non-randomized controlled experimental study (Humans)	Two healthy subjects	1. FS (STL/PLY) 2. IOS (STL) 3. DS + virtual facebow fork 4. CBCT + implant planning software (STL) 5. Open-source software 6. CAD software	1. Application (Hege 3D scanner;) in (iPhone 11 Pro; Apple Inc., Cupertino, CA, USA) 2. (TRIOS Color Pod; 3Shape, Copenhagen, Denmark) 3. (D2000; 3Shape, Copenhagen, Denmark) 4. (3D Accuitomo 170; J Morita) + (Blue Sky Plan; Blue Sky Bio, LLC, Libertyville, IL, USA) 5. (Blender 3D; Blender Foundation, Amsterdam, Netherlands) 6. (Exocad version 2.2; exocad GmbH, Darmstadt, Germany)	1. Facial information 2. Maxillary dentition 3. Alignment of dentition and the face 4. Model with face soft tissue and maxillary dentition 5 + 6. A composite model including the face, facebow fork, and the maxillary dentition	CBCT + FS + IOS: Surface-based + the best-fitting algorithm	Prosthetic dentistry: CAD/CAM	1. Feasibility 2. Accuracy of the VF: (1) High trueness: 1.14 ± 0.40 mm (2). High precision: 1.08 ± 0.52 mm (the difference of eight measurements was small)	The factors influencing accuracy were not explored.
Revilla-Leon M [23]/2022	Case report: method description (Humans)	One subject	1. IOS(STL) 2. FS (PLY+STL) 3. Extra-oral scan body system 4. DS (STL) 5. Open-source software 6. 3D modeling software 7. Avatar generator software (OBJ) 8. Motion Engine software program + a facial tracking app in a smartphone	1. (TRIOS 4, wireless, v. 21.2.0; 3Shape A/S, Copenhagen, Denmark) 2. (Instarisa facial scanner; Instarisa, Clovis, CA, USA) 3. (AFT Scan Body Teeth; AFT Dental System) 4. (T710 Scanner; Medit Inc., Seoul, Republic of Korea) 5. (Blender 3D; Blender Foundation, Amsterdam, Netherlands) 6. (Zbrush 2022; Maxon, Friedrichsdorf, Germany) 7. (Character Creator, v.3.44; Reallusion, SFO, CA, USA) 8. (Iclone, v.7.93; Reallusion) + (Live Face, v.1.08; Reallusion) in (iPhone 12 Pro; Apple Inc., Cupertino, CA, USA)	1. Dentitions and a maxillo-mandibular registration 2. Facial scan with and without scan body system 3. Registration tool 4. Digitizing scan body 5. All data converted to the OBJ format 6. Avatar without the hair and eyes 7. Avatar with the hair and eyes, dentition, and scan body 8. Virtual patient with lip dynamics	IOS + DS: fiducial markers based (extra-oral scan body system)	1. Prosthetic dentistry 2. Maxillofacial surgery	Feasibility	1. Lack of validation of accuracy 2. Complicated and time-consuming
Lepidi L [24]/2021	Case report: method description (Humans)	One subject	1. IOS(STL) 2. FS(OBJ) 3. DS(STL) 4. CBCT (DICOM) 5. Dental CAD software (STL)	1. (Cs 3600, Carestream Health, Rochester, NY, USA) 2. (Bellus 3D; Bellus 3D Inc., Campbell, CA, USA) 3. (Neway, Faro Technologies Inc., Brescia, Italy) 4. (Cs 9600 3D, Carestream Health, Rochester, NY, USA) 5. (Exocad; exocad GmbH, Darmstadt, Germany)	1. Maxillary and mandibular fixed prosthesis in the desired CR and VDO 2. Facial scan with and without the fork 3. Cast with implant analogs 4. CBCT image contained the restorations 5. Virtual patient mounting to the VA The occlusal discrepancy between MICP/CR	1. Cast with implant analogs, the casts with prostheses, and FS (DS + IOS + FS): fiducial markers based 2. FS with and without fork (FS + FS): point-based	Prosthetic dentistry	Feasibility	1. Lack of validation of accuracy 2. Complicated and time-consuming
Kim SH [25]/2020	Non-randomized controlled experimental study (phantom)	A skull phantom	1. MDCT + occlusal splint with a registration body 2. DS 3. The flat-panel display monitor with a 3D depth camera 4. EM tracking system	1. (SOMATOM Sensation 10, Siemens, Munich, Germany) 2. (Maestro 3D, Maestro, Pisa, Italy) 3. (QCT130, One Inc., Seoul, Republic of Korea) 4. (Aurora, Northern Digital Inc., Waterloo, ON, Canada)	1. The maxillary and skeletal models 2. The maxillary dentition 3. 3D virtual objects superimposed on the real patient image 4. 3D tracking positions of the bone segment and the reference in physical space	1. EM tracking + CT image spaces + AR camera spaces: fiducial markers based (markers on the registration body) 2. Dentition + maxillary model (MDCT + DS): ICP algorithm	Orthognathic surgery: AR-assisted free-hand orthognathic surgery	High accuracy: the MADs of the difference between actual and measured positions exhibited no significant differences between the SRT (0.20, 0.34, 0.29) and BRT (0.23, 0.37, 0.30)	Lack of validation of usability and accuracy in real patients
Lee SJ [26]/2019	Non-randomized controlled experimental study (phantom and humans)	A skull phantom + a patient	1. MDCT + Occlusal splint attached with a registration body 2. DS 3. EM tracking system	1. (SOMATOM Sensation 10, Siemens, Munich, Germany) 2. (Maestro 3D, Maestro, Pisa, Italy) 3. (Aurora, Northern Digital Inc., Waterloo, ON, Canada)	1. MPS and other skeletal models 2. Artifact-free model of the dentitions + final occlusal model 3. 3D tracking positions of MPS in a virtual maxillomandibular complex	1. CT image + physical spaces: fiducial markers based (registration body) 2. Dentition + maxillary model (MDCT + DS): ICP algorithm	Orthognathic surgery: (MPS repositioning; model-guided surgery)	1. Convenient 2. Accuracy: (The RMS differences between the simulated and intraoperative MPS models and between the simulated and postoperative CT models were 1.71 ± 0.63 mm and 1.89 ± 0.22 mm, respectively.)	Further development is needed to increase accuracy by reducing technical errors in the tracking devices, imaging errors from the modalities, registration errors, application errors, and human error
Li J [27]/2020	Case report: case Reports (humans)	One subject	1. IOS(STL) + a wax rim 2. CBCT (DICOM) + gothic arch 3. FS (STL) 4. Implant planning software 5. Dental CAD software + free CAD software.	1. (TRIOS; 3Shape, Copenhagen, Denmark) 2. (3D Accuitomo 170; J. Morita, Osaka, Japan) 3. (3dMDtrio System; 3dMD, Atlanta, GA, USA) 4. (Blue Sky Plan; Blue Sky Bio, LLC Libertyville, IL, USA). 5. (Exocad version 2.2; exocad GmbH, Darmstadt, Germany) + (Meshmixer; Autodesk, San Rafael, CA, USA)	1. Arches in approximate CR and VDO 2. (1) The maxilla, mandible, infraorbital points, and external acoustic meatus (2) The CR and VDO records 3. Smiling face 3D reconstruction of the face 4. Two-piece surgical templates 5. (1) Custom bases for a gothic arch tracer (2) The 3D images of FS, IOS, and skull mounted on a VA	Unknown	Dental implant surgery Prosthetic dentistry	Feasibility: successful prosthetic outcome	1. Obtaining an occlusal record with increased VDO using the gothic arch tracer is difficult for the patient with remaining teeth. 2. Lack of a quantitative validation of accuracy
Li J [28]/2021	Case report: method description (Humans)	Unknown	1. IOS (STL) 2. Dental CAD software 3. CBCT (DICOM) 4. FS (PLY) 5. Implant planning software (STL) 6. Open-source 3D software program	1. (TRIOS 3; 3Shape A/S, Copenhagen, Denmark) 2. (exocad; exocad GmbH, Darmstadt, Germany) 3. not mentioned 4. (Hege 3D scanner) in (iPhone 11 Pro; Apple Inc., Cupertino, CA, USA) 5. (BlueSkyPlan v4.70; Blue Sky Bio LLC, Libertyville, IL, USA) 6. (Blender 3D; Blender Foundation, Amsterdam, Netherlands)	1. Arches in approximate CR and VDO 2. Design of the tooth-supported template 3. The maxilla, mandible, infraorbital points, and external acoustic meatus 4. Facial scan 5. 3D bone model and face model 6. Tooth-supported gothic arch tracer Alignment of the facebow and the skull Virtual patient with FS, IOS, CBCT	Unknown	Dental implant surgery Prosthetic dentistry: (complex implant-supported Prostheses)	1. Predictability 2. Feasibility	1. The need for a CBCT scan 2. Lack of a quantitative validation of accuracy
Olszewski R [29] /2008	Non-randomized controlled experimental study (Humans)	Unknown	1. CT (DICOM) + MRI 2. DS 3. 3D tracking device 4. System for planning and assisting orthognathic surgery	Data integration module of the system: MedicalStudio1 platform 1. ACRO 3D: 3D CT based craniofacial cephalometric Analysis 2. ACROTooth: virtual occlusion 3. TMJSim: TMJ movement Simulation 4. ACROSim: virtual surgery planning 5. ACROGuide: intra-operative AR assistance	1. Maxilla, mandible, skull, and skin 2. Dental casts. 3. Mandibular movement: (1) translation and the rotation (2) the centric position for the condyles (3) The joint’s degrees of freedom 4. Virtual models	The physical space + the digital world: 1. algorithm (minimizing the mean square distance between the points) 2. fiducial markers (attached to the tracked surgical tools) based	Maxillofacial orthognathic surgery	Complete Accuracy: 1. ACRO 3D module: validated 2. TMJSim: partly validated; ACROGuide: partly validated.	Before clinical application: 1. accuracy needs to be further validated. 2. technology and algorithms need to be further improved.
Fushima K [18]/2007	Non-randomized controlled experimental study (Humans)	More than 50 cases	1. Lateral and PA cephalograms 2. 3D motion tracking 3. Facebow-transfer 4. Surgical simulation system: mandibular motion tracking system (ManMoS)	1. Scanner (ES-2200, Epson Co., Owa, Suwa, Nagano, Japan) 2. (Polhemus, Colchester, VT, USA) 3. Unknown 4. (FASTRAK, Virtual Realities, LLC., League City, TX, USA)+2	1. Skeletal in a virtual space 2. Mandibular motion tracking 3. A record of how the upper dentition relates to the TMJ 4. Pilot surgical prediction and real-time surgical simulation: model of the craniofacial skeleton with the centric stops of the dental arches	Dentition + skeleton + motion tracking in a virtual space: fiducial markers based	Maxillofacial orthognathic surgery	1. Feasibility 2. Trueness: sufficient (small SD) 3. Precision: SD in the 40 recordings was less than 0.1 mm	Complicated and time-consuming
Kois JC [30]/2022	Case report: method description (Humans)	Unknown	1. IOS 2. KFRG 3. Photography 4. CAD software 5. VA module	1. (TRIOS 3; 3Shape A/S, Copenhagen, Denmark) 2. (Kois Center, LLC, Seattle, WA, USA) 3. Digital single-lens reflex camera (D850; Nikon Inc., Tokyo, Japan) 4. (DentalCAD; exocad GmbH, Darmstadt, Germany)	1.Maxillary and mandibular arches 2. NHP 3. Photograph of dentition and face 4. Virtual Orientation: the project scene 5. Facially generated virtual mounting	IOS + photograph: “Align Mesh” tool	Prosthetic dentistry	Feasibility	Lack of the validation of accuracy
Kwon JH [31]/2019	Case report: method description (Humans)	One subject	1. CBCT (DICOM) 2. Image processing software (STL) 3. DS 4. FS + targets 5. Image registration tool	1. (PaXZenith3D, Vatech Co., Ltd., Hwaseong, Republic of Korea;) 2. (OnDemand3D, Cybermed Co., Ltd., Seoul, Republic of Korea.) 3. (Identica hybrid, Medit Inc., Seoul, Republic of Korea.) 4. (Rexcan CS2, Medit Inc., Seoul, Republic of Korea.) 5. (Ezscan8, Medit Inc., Seoul, Republic of Korea)	1. Maxilla and mandible 2. 3D skull and jaws 3. digital casts 4. The oral cavity and face in MICP 5. A 3D model with a CBCT scan Real-time mandibular motions	CBCT + DS: fiducial markers based (to generate the transformation matrix by comparing reference points)	Prosthetic dentistry	1. Convenient 2. Stability: 36 mm in the mandible 30.78 mm/37.74 mm in the left/right condyle 3. Accuracy: high (4.1–6.9 mm)	The sample size needs to be expanded for further validation.
Lam WYH [32]/2016	Non-randomized controlled experimental study (Humans)	Unknown	1. FS with and without facebow 2. DS 3. IOS 4. Open-source software 5. CBCT scan+ the occlusal wafer + radiopaque markers	1. (3dMDface; 3dMD Inc, Atlanta, GA, USA) 2. (Handyscan 3D; Creaform) 3. (True Definition; 3M ESPE, Iverson Dental Labs, MARB, CA, USA) 4.(MeshLab v1.3.3; Visual Computing Lab of the ISTICNR, Pisa, Italy) 5. Image analysis software (3D Slicer9 version 4.3; Slicer community, Boston, MA, USA)	1.3D face with/without facebow 2. Facebow 3. (1) Dentitions and the maxillomandibular relationship; (2) Buccal surface of the maxillary dentition and the occlusal wafer 4. Virtual patients transferred to the VA 5. The facial skin, teeth, and radiographic markers	1. FS + DS + IOS: point-based + ICP algorithm 2. Face-bow (DS) + CBCT: fiducial markers based + point-based algorithm	Prosthetic dentistry	1. Feasibility 2. Accuracy (1) error in tooth registration: less than 1 mm (2) the facial alignment (the mean distances of the nasion, alares, and tragions: 0.83 mm, 0.77 mm, and 1.70 mm)	1. The registration accuracy needs to be improved. 2. Distortion needs to be avoided. 3. Time-consuming
Lam WYH [33]/2018	Non-randomized controlled experimental study (Humans)	One subject	1. FS 2. IOS 3. DS 4. Open-source software 5. CBCT 6. CAD software	1. (3dMDface; 3dMD Inc., Atlanta, GA, USA) 2. (True definition scanner; 3M ESPE, Iverson Dental Labs, MARB, CA, USA) 3. (DAVID SLS-3; Hewlett-Packard, Palo Alto, CA, USA) 4. (MeshLab v1.3.3; Visual Computing Lab of the ISTI-CNR, Pisa, Italy) 5. (ProMax 3D Mid, Planmeca OY, Helsinki, Finland) 6. (Exocad; Exocad GmbH, Darmstadt, Germany)	1. 3D face in NHP position or with a VF 2. (1) Dentitions and the maxillomandibular relationship; (2) Buccal relationship of the maxillary teeth and VF 3. VF 4. The dentition and the 3D facial photographs in NHP 5. The dentition, jaws, and 3D facial photograph in NHP 6. Virtual patients transferred to the VA	FS + DS + IOS: point-based + ICP algorithm Face-bow (DS) + CBCT: fiducial markers based + point-based algorithm	Prosthetic dentistry	Good precision of the SP NHP technique: positional differences of less than 1 degree and 1 mm in five repeated measurements in one patient	The sample size needs to be expanded for further validation.
Shao J [34]/2019	Case report: clinical report	One subject	1. Multi-slice spiral CT 2. Facial photograph 3. Imaging software 4. DS 5. CAD software 6. CAD software + 3D printer	1. (Philips MX16 EVO CT; Koninklijke Philips N.V., Amsterdam, NL) 2. (3dMDface System; 3dMD, Atlanta, GA, USA) 3. (Dolphin Imaging & Management Solutions; Patterson Dental, Chatsworth, CA, USA) 4. (3ShapeA/S, Copenhagen, Denmark) 5. (Exocad GmbH, Darmstadt, Germany) 6. (Formlabs Form 2; Formlabs, Boston, MA, USA)	1. Bone and dentition 2. Virtual face model with bone and soft tissue in real-time in NHP 3. Simulation of the lateral facial profile influenced by the retrusion of anterior maxillary teeth Dentition casts scan preserving their articulator-mounted relationship 5. The restorations with incisal edges were retruded for 5.0 mm 6. Prosthetically driven planning	Unknown	Dental implant surgery Prosthetic dentistry	Feasibility	Lack of the validation of accuracy
He S [35] /2016	Non-randomized controlled experimental study: (Humans)	One subject	1. CBCT + GALAXIS 3D software (DICOM) 2. SICAT JMT + system	1. (Sirona Galileos, Bensheim, Germany) 2. (SICAT Function; SICAT, Bonn, Germany)	1. CT images: condylar status 2. (1) Jaw movements + incisor ranges (2) Models integrating CBCT and JMT data: movement of the mandible (including the translation of the condyles)	CBCT + JMT data: fiducial markers based (radiopaque markers on bite tray)	Maxillofacial orthognathic surgery Digital dentistry	1. Reliable accuracy: the same positions between the simulated condylar position with that in the second CBCT 2. High precision	Further studies are needed to validate its accuracy.
Park JH [36] /2021	Case report: clinical report (Humans)	One subject	1. IOS + software program (STL) 2. CBCT 3. VA program (STL)	1. (TRIOS; 3Shape, Copenhagen, Denmark) + (Ortho Analyzer; 3Shape, Copenhagen, Denmark) 2. (Alphard Vega; Asahi Roentgen, Kyoto, Japan) 3. (R2GATE 2.0.0; Megagen, Seoul, Republic of Korea)	1. Cast in CRO and MICP position 2. CBCT model in CRO 3. (1) The maxillary cast registered on the CBCT model (2) VA (3) Models Superimposing the mandible position in CRO and MICP	IOS + CBCT: point-based + ICP algorithm	Prosthetic dentistry	Feasibility	Lack of the validation of accuracy
Amezua X [37] /2021	Non-randomized controlled experimental study (phantom)	A skull phantom	1. CAD software 2. DS by Industrial reference scanner (STL) 3. FS by the reference scanner (STL)/SWL scanner/SFM scanner methods (OBJ) 4. RE software	1. (Solid Edge ST10; Siemens, Munich Ger) 2. (ATOS Compact Scan 5M scanner with ATOS Professional V7.5 software; GOM, GmbH, ZEISS, BS, Ger) 3. SFM: (PENTAX K-S1; Ricoh Imaging Co, Ltd., Tokyo, Japan) + (Agisoft Metashape Professional; Agisoft, SPB, Russia) SWL: (Go! SCAN20 scanner with VX element 6.3 SR1 software; Creaform, Inc., Lévis, CAN) 4. (Geomagic Studio 2013; Geomagic, Inc., RTP, NC, USA)	1. IOTE 2 + 4. The maxillary model without regions not correspond to the teeth 3. (1) FS with IOTE (2) IOTE-free FS (3) FS with the mouth open 4. Models aligning IOTE-free FS and the maxillary scan	FS + DS/transferring maxillary digital scan to standard virtual patient: point-based + ICP algorithm	Prosthetic dentistry	Reliable accuracy: (below 1 mm): 0.182 mm for the RE group, 0.241 mm for the SWL group, and 0.739 mm for the SFM group	1. Further studies are needed to validate its accuracy. 2. In vitro experiments may underestimate the scanning error.
Dai F [38] /2016	Case report	One subject	1. DS (STL) 2. Spiral CT (DICOM) + 3D software (STL) 3. Ultrasonic axiograph Arcus Digma system (TXT file) 4. RE software 5. Mathematical software MATLAB 7.0 + Amira software FE modeling 6. Analysis software Ansys 15.0	1. (Roland DG., Hamamatsu, Japan) 2. (PHILIPS Inc., Andover, MA, USA) + Amira5.2.2 (Visage Imaging Inc., SD, CA, USA) 3. (KaVo, Biberach, Germany) 4. Rapidform 2006 (Inus Technology Inc., Seoul, Republic of Korea) 5. MATLAB 7.0 (Math Works Inc., Natick, MA, USA) 6. (ANSYS Inc., Canonsburg, PA, USA)	1. The upper cast with the bite fork and the occlusion 2. 3D bone, muscle, and teeth 3. Mandibular movement 4. Static model of the masticatory system 5. Dynamic model with a simulation of mandibular movement 6. The FE masticatory system model	1. Casts + casts made in occlusion (IOS + IOS): regional registration method 2. Registration of the different coordinate systems: based on the global coordinates + the (bite fork)	Digital dentistry Orthodontics	1. Feasibility 2. Accuracy: (1) the static masticatory system model: small difference (0.32 ± 0.25 mm) indicated good accuracy (2) the FE model showed accuracy similarity to that of the T-Scan (3) The accuracy of the 3D Arcus Digma system: 0.1 mm and 1.5°	The sample size needs to be expanded for further validation.
Savoldelli C [39]/2012	Non-randomized controlled experimental study: (Humans)	One subject	1. Multislice CT with a splint 2. MRI with a splint 3. 3D image segmentation software 4. FE analysis software	1. (General Electric Medical System, UWM, WI, USA) 2. Gyroscan Intera 1.5-T MR system (Philips Medical Systems, Best, NL) 3. (AMIRA^®^) (Visage Imaging, Inc., SD, CA, USA) 4. FORGE (Transvalor, Glpre 2005, Antibes, France)	1. Bone components of skull and mandible, dental arches when the jaw was opened 10 mm. 2. Soft tissues such as joint discs, temporomandibular capsules, and ligaments when the jaw was opened 10 mm. 3. Virtual models with surface and volume meshes of the above components 4. (1) Boundary conditions for closing jaw simulations by different jaw muscles (2) Stress distribution in both joint discs	MRI + CT: based on the anatomical structures (Hounsfield unit values + manual identification)	Digital dentistry Orthodontics	1. Feasibility 2. Accuracy (high): stress levels (5.1 MPa) were within the range of reported stress (0.85–9.9 MPa)	1. The material behavior of the articular discs was a linear elastic model and not a non-linear material model. 2. the sample size needs to be expanded for further validation.
Terajima M [4]/2008	Case report: method description (Humans)	One subject	1. CT + image processing software + visualization software 2. DS (VIVID format) 3. Jaw-movement analyzer 4. Image measurement software	1. CT scanner (Aquilion, Toshiba Medical, Tokyo, Japan) + (Mimics version 7.0, CDI, Tokyo, Japan) + (Magics, CDI, Tokyo, Japan) 2. (VIVID 900, Minolta, Tokyo, Japan) 3. (TRI-MET, Tokyo-Shizaisha, Tokyo, Japan) 4. (3D-Rugle, Medic Engineering, Kyoto, Japan)	1. Reconstruction of images integrating the CT, the 3D dental surface, ceramic spheres 2. Dental surface + ceramic spheres 3. Mandibular movement 4. Condyle position relative to the condylar fossa + contact areas during jaw movements	1. CT +DS: fiducial markers based (ceramic balls) 2. Registration of 3D maxillofacial-dental images and that in the TRI-MET system: the least squares method	Digital dentistry Orthodontics Orthognathic surgery	Feasibility	Further studies are needed to validate its accuracy.
Perez-Giugovaz MG [40] /2021	Case report: method description (Humans)	One subject	1. IOS (STL) 2. CAD software (STL) 3. Printer software 4. FS + a facebow record 5. DS (STL) 6. CAD software	1. (Cs 3600, Carestream Health, Rochester, NY, USA) 2. (MeshMixer; Autodesk, SR, CA, USA) 3. (ChiTuBox V1.7.0; ChiTuBox, Shenzhen, China) 4. (Bellus Face Camera Pro; Bellus3D Inc., Campbell, CA, USA) 5. (Open Technologies Small; Faro, Lake Mary, FL, USA) 6. (Dental CAD Plovdiv; exocad GmbH, Darmstadt, Germany)	1. Maxillary and mandibular casts 2. Virtual design of: (1) custom tray and mandibular occlusion rim with gothic arch tracer (2) scan body 3. Manufacture of the above devices 4. Facial scan with occlusion rim and scan body 5. Casts with occlusion rims and the scan body 6. The virtual patient with the casts mounted on the VA	1. FS + scan body (DS): fiducial markers based 2. FS with occlusion rim + FS with scan body: facial point-based	Digital dentistry Prosthetic dentistry: CAD/CAM	Feasibility	Further studies are needed to validate its accuracy.
Solaberrieta E [41]/2015	Case report: method description (Humans)	Unknown	1. IOS 2. Camera + reverse engineering software + target 3. Reverse engineering software	1. (3Shape TRIOS; 3Shape A/S, Copenhagen, Denmark) 2. (Nikon D3200; Nikon Inc., Tokyo, Japan) + (Agisoft Photoscan; Agisoft LLC, SPB, Russia) 3. Rapidform 2006 (Inus Technology Inc., Seoul, Republic of Korea)	1. (1) Maxillary and mandibular casts (2) The casts + facebow fork (3) Casts in the VA in MICP 2. 3D face with targets on the facebow fork 3. (1) Alignment of the face and facebow fork, the maxillary cast, and the facebow fork (2) Casts transferred to VA	1. The maxillary cast + facebow fork + 3D face-facebow fork (IOS + FS): best-fit command 2. Alignment of cranial coordinate system: facial point-based	Digital dentistry Prosthetic dentistry: CAD/CAM	Feasibility	Additional studies need to validate the accuracy of the new systems.
Granata S [42]/2020	Case report (Humans)	One subject	1. Geometric occlusal registration prototype device 2. FS (OBJ) 3. IOS (PLY/STL) 4. CBCT with DGB1 (DCM) 5. DS 6. 3D-guided surgery planning software + CAD design software	1. (DGB) (Digitalbite; Digitalsmile srl, Pietracamela, Italy) 2. (Bellus3D; Bellus3D Inc., Campbell, CA, USA) 3. (Cs 3600, Carestream Health, Rochester, NY, USA) 4. (Cs 9300, Carestream Health, Rochester, NY, USA) 5. (InEosXs; Dentsply Sirona, Charlotte, NC, USA) 6. (DDS-Pro; Dentalica Spa, Milano, Italy) + (Exocad; Exocad GmbH, Darmstadt, Germany)	1. Auxiliary equipment for registration 2. (1) Face with a maximum smile, face with DGB1, and face with mouth open and MICP (2) Face with DGB2 in three poses 3. Maxillary and mandibular dental arches 4. Bone and dental arches after placing DGB1 5. DGB1 devices 6. Virtual patient and virtual prosthetic planning	FS in three poses + IOS: fiducial markers + geometric reference-based (DGB with radiopaque landmark) + best-fitting algorithm	Digital dentistry Prosthetic dentistry: CAD/CAM Dental implant surgery	Feasibility Dynamic Inexpensive	1. The distortion caused by the processing of the original files and the matching method 2. Further studies are needed to validate its accuracy.
Noguchi N [43]/2007	Case report (Humans)	One subject	1. DS + FS 2. Cephalometry + digital radiograph system 3. A digitizer 4. 3D shape analysis software	1. (SURFLACER 3D-VMS250/300, UNISN Inc., Osaka, Japan) 2. (FCR, Fuji Film Co., Ltd., Tokyo, Japan) 3. (KW4610, Graphtec, Yokohama, Japan) 4. (SURFLACER 3D-VMP300 (UNISN Inc., Osaka, Japan) 5. (Imageware Surfacer, Metrix Software Solutions Ltd., Montreal, Canada)	1. (1) Dentition and occlusal impression; (2) facial soft-tissue 2. Data for the mandible 3. The traced bone, teeth, and soft tissue 4. (1) Virtual models integrating the data above (2) Movement displayed using a color map	Projection-matching technique: based on the contour line of the projection image	Orthodontics Orthognathic surgery	1. Feasibility 2. Accuracy: registration error was the same as that in conventional Cephalometry.	Further studies are needed to validate its accuracy.

Abbreviations: cone beam computed tomography (CBCT); intraoral scanners (IOS); desktop scanners (DS); facial scanners (FS); virtual facebows (VF); virtual articulator (VA); standard tessellation language (STL); the object code (OBJ); polygon (PLY); digital imaging and communications in medicine (DICOM); data communication module (DCM); moving picture experts group 4 (MP4); text file (TXT); the iterative closest point (ICP) algorithm; computer-assisted design (CAD); maximal intercuspation position (MICP); centric relation (CR); vertical dimension of occlusion (VDO); natural head position (NHP); Kois Facial Reference Glasses (KFRG); electromagnetic (EM); mean absolute deviations (MADs); the root mean square (RMS); standard deviations (SD); mandibular proximal segment (MPS); intraoral transfer element (IOTE); reverse engineering (RE) software; finite element (FE) software.

**Table 3 jfb-14-00033-t003:** Summary of the dynamic data.

Type of Method	Reference	Ways to Acquire Dynamic Data	Type of Dynamic Data
Target tracking video	[20]	Target tracking video-camera of a mobile phone with 4000-pixel (4K) resolution (iPhone 7; Apple Inc., Cupertino, CA, USA)	Mandibular movements: mandibular kinematic path and TMJ kinematic path
FS + targets	[21,31]	FS (Rexcan CS2, Medit Inc., Seoul, Republic of Korea) + lip and cheek retractor + nonreflective targets attached to incisors	Real-time mandibular motions
JMT (Jaw motion tracker)	[25,26]	EM tracking system (Aurora, Northern Digital Inc., Waterloo, ON, Canada) + skin-attached dynamic reference frame	3D tracking of the positions of the bone segment
	[18]	EM tracking (Polhemus, Burlington, VT, USA) Facebow-transfer Three rectangular coordinate systems (Cartesian)	Mandibular motion tracking Real-time surgical simulation
	[4]	Optoelectronic analysis system with 6 degrees of freedom (TRI-MET, Tokyo-Shizaisha, Tokyo, Japan) Image measurement software (3D-Rugle, Medic Engineering, Kyoto, Japan)	4D display of mandibular movement Condyle position relative to the condylar fossa Contact areas during jaw movements
	[29]	Ultrasonic tracking device with six degrees of freedom (TMJSim: TMJ movement) in (MedicalStudio1 platform)	Mandibular movement: translation and the rotation The centric position for both of the condyles Joint’s degrees of freedom
	[35]	SICAT JMT + system (SICAT Function; SICAT, Bonn, Germany) (with ultrasonic tracking device)	Mandibular movements: opening, right and left lateral movement, and protrusion Incisor ranges movement of the mandible, including the translation of the condyles
	[38]	Ultrasonic axiograph Arcus Digma system (KaVo, Biberach, Biberach, Germany)	Mandibular movements
Facial tracking system	[23]	Facial tracking app (Live Face, v.1.08; Reallusion) in a smartphone (iPhone 12 Pro; Apple Inc., Cupertino, CA, USA)	Lip dynamics (including rest/“m” sound/smile/speech)
	[42]	FS (Bellus3D; Bellus3D Inc, Campbell, CA, USA) Geometric occlusal registration prototype device (DGB) (Digitalbite; Digitalsmile srl, Pietracamela, Italy)	Face with a maximum smile, with mouth open and MICP
	[34]	3D facial photograph Dolphin 3D Imaging	Lateral facial profile influenced by retrusion of anterior maxillary teeth.
	[43]	FS (SURFLACER 3D-VMS300, UNISN Inc., Osaka, Japan) Frontal and lateral cephalometry (FCR, Fuji Film Co. Ltd., Tokyo, Japan) Digitizer (KW4610, Graphtec, Yokohama, Japan)	The traced bone, teeth, and outline of the soft tissue
Method to acquire positional relationships of jaws, skull, and TMJ	[22,30]	FS (InstaRisa Facial Scanner; InstaRisa, Clovis, CA, USA)/photograph Scan body system (Kois Scan Body System; Kois Center, LLC, Seattle, WA, USA) VA (Panadent Articulator; Panadent, Colton, CA, USA) CAD software	Dynamic facial information NHP Maxillomandibular registration and jaw-correction function
	[41]	Photograph + reverse engineering software (Agisoft Photoscan; Agisoft LLC, SPB, Russia) + targets VF + VA Reverse engineering software	Mandibular position in MICP 3D face Maxillary and mandibular dentiton transfered on VA
	[19,42]	FS: smartphone (iPhone 11 Pro; Apple Inc., Cupertino, CA, USA) with a 3D scan application (Hege 3D scanner)/((Bellus3D; Bellus 3D Inc., Campbell, CA, USA) DGB) VF fork CAD software	Facial information Occlusion + position of the maxilla
	[24]	FS (Bellus 3D; Bellus 3D Inc., Campbell, CA, USA) VF CAD software	Position of the maxilla CR position with the joint axis of the VA Occlusal discrepancy
	[32,33]	FS (3dMDface; 3dMD Inc, Atlanta, GA, USA)/(DAVID SLS-3; Hewlett-Packard, Palo Alto, CA, USA) VF CAD software	The digital teeth to the 3D facial photographs in NHP Maxillomandibular relationship with maxilla
	[37]	FS by the reference scanner, SWL scanner, and SFM scanner RE software program	Maxillomandibular relationship with the maxilla Face with the mouth open
	[40]	FS (Bellus3D; Bellus3D Inc., Campbell, CA, USA) Facebow record CAD software program	The definitive casts mounted on the VA to simulate the jaw position
	[27,28]	FS: (3dMDtrio System; 3dMD, Atlanta, GA, USA)/application (Hege 3D scanner) in (iPhone 11 Pro; Apple Inc., Cupertino, CA, USA) CBCT + gothic arch tracing + articulator CAD software, Implant planning software	Maxillary and the mandible arches aligned in a proximal CR and VDO 3D face
	[36]	CBCT VF VA (R2GATE 2.0.0; Megagen, Seoul, Republic of Korea)	Mandible position in both CRO and MICP
FE analysis system	[38]	FE modeling and analysis software Ansys 15.0 (ANSYS Inc., Canonsburg, PA, USA) Reverse engineering software (Rapidform 2006 (Inus Technology Inc., Seoul, Republic of Korea))	A dynamic model of the individualized masticatory system including the cranio-maxilla, the mandible, masticatory muscles, and 28 complete teeth The FE masticatory system model
	[39]	FE analysis software (FORGE (Transvalor, Glpre 2005, Antibes, France)) 3D image segmentation software (AMIRA^®^) (Visage Imaging, In, SD, CA, USA)	Boundary conditions for closing jaw simulations by different load directions of jaws muscles The stress distribution in both joint discs during closing conditions

Abbreviations are the same as that of Table 2.

**Table 4 jfb-14-00033-t004:** Summary of the software to create 4D patients.

References	Software	Type	Possible Registration	Imported Data
[19,20,23,28]	(Blender 3D; Blender Foundation, Amsterdam, The Netherlands)	Free open-source CAD software	Point-based Direct linear transform (DLT) algorithm	CBCT (DICOM) IOS (STL) JMT video (MP4) FS (PLY + STL)
[19,21,22,24,27,28,30,34,40,42]	(Exocad; exocad GmbH, Darmstadt, Germany)	CAD software	Point-based Fiducial markers-based Surface-based Best-fitting algorithm	CBCT (DICOM) IOS/DS (STL/PLY) FS (STL/PLY/OBJ) Photograph
[23]	(Zbrush 2022; Maxon, Friedrichsdorf, Ger-many)	3D modeling software	Fiducial markers-based	IOS/DS (OBJ)
[19,21]	(EzScan8, Medit Inc., Seoul, Republic of Korea)	The image registration software	Point-based Horn alignment algorithm ICP algorithm	CBCT (STL transferred from DICOM) DS (STL) FS (STL)
[32,33]	(MeshLab v1.3.3; Visual Computing Lab of the ISTICNR, Pisa, Italy)	Open-source software	Point-based Fiducial markers based ICP algorithm	CBCT, IOS, FS
[34]	(Dolphin Imaging & Management Solutions; Patterson Dental, Chatsworth, CA, USA)	3D Imaging software	Unknown	CT Facial photograph DS
[36]	(R2GATE 2.0.0; Megagen, Seoul, Republic of Korea)	Virtual articulator program	Point-based ICP algorithm	CBCT (DICOM) IOS (STL)
[38]	Rapidform 2006 (Inus Technology Inc., Seoul, Republic of Korea)	Reverse engineering software	Regional registration method Global coordinates-based	CT (DICOM) DS (VIVID) 3D motion tracking (TXT)
[38]	(ANSYS Inc., Canonsburg, PA, USA)	FE analysis software	*	*
[39]	(AMIRA) (Visage Imaging, Inc., SD, CA, USA)	3D Imaging software	Hounsfield-unit values-based Manually identification of anatomical structures	CT, MRI
[39]	FORGE (Transvalor, Glpre 2005, Antibes, France)	FE analysis software	*	*
[41]	Rapidform 2006 (Inus Technology Inc., Seoul, Republic of Korea)	RE software	Best-fit command Facial point-based	IOS 2D photograph
[25,26]	(Aurora, Northern Digital Inc., Waterloo, ON, Canada)	Orthognathic navigation systems	Fiducial markers based ICP algorithm	CBCT/MDCT (DICOM) DS (STL) IOS (STL)
[29]	MedicalStudio1 platform: ACRO	Orthognathic planning and navigation systems	Fiducial markers based Algorithm minimizing the mean square distance between the points	CT (DICOM) MRI DS
[18]	ManMoS: (FASTRAK, Virtual Realities, LLC., League City, TX, USA) + (Polhemus, Colches-ter, VT, USA)	Orthognathic simulation systems	Fiducial markers based	Lateral and posteroanterior cephalograms 3D motion tracking
[35]	(SICAT, Bonn, Germany)	Orthognathic planning systems	Fiducial markers based	CBCT (DICOM) 3D motion tracking
[4]	(TRI-MET, Tokyo-Shizaisha, Tokyo, Japan)	Orthognathic planning systems	Fiducial markers based The least squares method	CT DS 3D motion tracking
[43]	(Imageware Surfacer, Metrix Software Solutions Ltd., Montreal, Canada)	Orthognathic planning systems	Projection-matching technique	Lateral and posteroanterior cephalograms DS FS 3D motion tracking

* The model of the FE software is based on the 3D models from other software, so its types of information and registration are not shown in the table. Abbreviations are the same as that of Table 2.

## Data Availability

Not applicable.

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
