# Peer review of "Four-Dimensional Superimposition Techniques to Compose Dental Dynamic Virtual Patients: A Systematic Review"

_jfb, 2023, doi:10.3390/jfb14010033_

Round 1

Reviewer 1 Report

Four-dimensional superimposition techniques to compose dental dynamic virtual patients: A systematic review

Registration and protocol for a systematic review are mandatory, not “not applicable” as stated in the PRISMA checklist.

This paper overall is interesting; however, it is very difficult to follow.

The authors should focus more on the clarity of the information, rather than on the quantity.

Why were searchers performed only between 2007-03/2022-11?

The snowball technique of article selection is not very well explained, although it appears in

Figure S1.

The authors did not relate to the aim of the paper, mostly due to new information added, especially in the discussion section (see table 4)

The number of the excluded reason in the prisma flowchart should be explained

3.4 outcomes are difficult to follow – page 24 lines 5-46 page 25 47-156

ST+D+OA+MM/JP

SK+ST+D+JP

SK+ST+D+MM

SK+D+MM+TMJ soft tissues

SK+D+MM

ST/SK+ D+ JP

Should be better explained and there should be an abbreviation list

The applications of 4D virtual patients in clinical practice should be better highlighted.

The clinical relevance should be emphasized

Limitations should be provided

The conclusion should be rewritten based on the results, it is unclear.

Author Response

Point 1: Registration and protocol for a systematic review are mandatory, not “not applicable” as stated in the PRISMA checklist. 

Response 1: We thank the reviewer for pointing out this issue. Indeed, registration before formal screening can avoid duplication and ensure meaningful studies. Before and after the research, we looked up topics related to "4D virtual patient" on PROSPERO and did not find similar studies. Most of the articles in our study were preclinical trials or case reports, which did not fit well with Prisma's principle of collecting interventional RCTs. In addition, this study belongs to the literature reviews that use a systematic search, which PROSPERO does not accept. Therefore, we acknowledge that the present review was not registered, and there was no protocol prepared. We have stated this in the paragraph of subsection 2.1 Eligibility criteria in the manuscript and modified the "not applicable" stated in the PRISMA checklist. The present review was not registered because it belongs to the literature reviews that use a systematic search, which PROSPERO does not accept. (Page2 Lines85-87)

Point 2: This paper overall is interesting; however, it is very difficult to follow.

The authors should focus more on the clarity of the information, rather than on the quantity.

Response 2: Thank you for your comment and suggestions. In the revised manuscript, we mainly concentrated on three information sections, including the dynamic data collection methods of 4D virtual patients, 4D superimposition techniques, and clinical applications of 4D virtual patients. We partially reconstructed the result (3.4.1, 3.4.2, 3.4.3), discussion (4.1, 4.2, 4.3), and conclusion (first three points) and elaborated the information and findings of the above three sections accordingly, hoping to enhance the paper's readability.

In some paragraphs, we enrich the information of the article, such as 4.3 Applications of 4D virtual patients. In some sections, we may begin by stating the number of similar studies for easier categorization, and then we give a detailed description of similar dynamic information/techniques/applications.

Point 3: Why were searchers performed only between 2007-03/2022-11?

Response 3: Thank you for your question. The searches were performed up to November 2022. The first search was performed on August 5, 2022, and Weekly literature tracking was conducted separately to obtain the latest relevant literature. We corrected the ambiguity in our description. (P1 lines15-16) The included publications were dated from 2007-03 to 2022-11, without intentional time restriction, since the 4D virtual patient is a new technology recently proposed. (P4 lines 171-173)

Point 4: The snowball technique of article selection is not very well explained, although it appears in Figure S1.

Response 4: As the reviewer suggested, we added the additional explanation to the “snowball procedure” in the paragraph of subsection 2.2 Information sources of the revised manuscript. The "snowball procedure" is a multi-round forward screening, after the full-text screening, to search the eligible papers from the reference lists of the included papers. Once a new study/reference is included, its references are called snowball papers, which will undergo a new round of snowball screening. This procedure ends only when no snowball papers can be included in the last round [13]. (P2 lines103-107)”

Point 5: The authors did not relate to the aim of the paper, mostly due to new information added, especially in the discussion section (see table 4)

Response 5: The reviewer’s question that no new information and tables should be added to the discussion was very enlightening. The software-related explanation (4.2 Software programs to create the virtual patient) is one of the essential parts of the second significant section "4D superimposition techniques", but it should not appear in the discussion only. Now, this part is placed in the RESULT in the revised manuscript. (P27 line 190, P28 lines238-275) Also, we tried our best to ensure that the debate corresponded to the article's result and purpose.

Point 6: The number of the excluded reason in the Prisma flowchart should be explained

Response 6: Thank you for your suggestions. Following the reviewer's advice, we state the total number of excluded reasons in parentheses after “excluded reasons” in the Prisma flowchart, followed by the excluded reasons. (Figure1) The number of excluded reasons exceeds that of excluded articles only shows up at the screening phase, where the total number of reasons excluded is 1344, Still, only 1258 articles are excluded because some articles have more than two reasons excluded. We added an explanation in the figure caption of Figure 1 in the revised manuscript. The number of excluded reasons exceeds that of excluded articles only at the screening phase, where the total number of reasons excluded was 1344, but 1258 articles were excluded. (P6 lines222-224)”

Point 7: 3.4 outcomes are difficult to follow – page 24 lines 5-46 page 25 47-156

ST+D+OA+MM/JP

SK+ST+D+JP

SK+ST+D+MM

SK+D+MM+TMJ soft tissues

SK+D+MM

ST/SK+ D+ JP

Should be better explained and there should be an abbreviation list

Response 7: To better explain 3.4 outcomes, we added a summary of the main content of 3.4.2 and the acronym's meaning in front of this part. (P26 lines91-97) We also added an abbreviation list as a back matter. (P34 lines599-613) We sincerely hope our modifications will increase the readability of this paragraph.

3.4 outcomes: The main components of stomatognathic information include static information such as the skeletal components (SK) of the skull, jaws, and TMJ, dentition (DENT), dynamic information such as mandibular movement (MM), jaw position (JP), occlusal analyses (OA), and motion of TMJ soft tissues and muscles. Moreover, soft tissues of the face (SF) can be either in a dynamic or static form. Three to five of these components were combined to create virtual models based on 4D superimposition techniques. These models are created using various software systems.

In 3.4 outcomes, we focused on the three main parts: “3.4.1 dynamic data collection methods of 4D virtual patients, 3.4.2 4D superimposition techniques, and 3.4.3 clinical applications of 4D virtual patients.” Since most studies have commonalities in dynamic data collection methods and clinical applications, we summarized them in 3.4.1 and 3.4.3. Since the technology involved in 4D superimposition techniques is complex, first of all, according to the classification of information, we divided them into single studies to discuss what method was used to obtain what information during the construction of a 4D virtual patient. The superimposition technique and the final feasibility/accuracy were also presented. Since there are also commonalities among the software involved in the 4D superimposition techniques, we summarized it at the end of 3.4.2. (P27 line 190, P28 lines238-275)

Point 8: The applications of 4D virtual patients in clinical practice should be better highlighted.

The clinical relevance should be emphasized

Response 8: Thank you for your kind suggestion. Following your advice, we have added more information about the applications of 4D virtual patients in clinical practice, including prosthetic dentistry, dental implant surgery, maxillofacial surgery, and orthodontics. (P30 lines 281-311, P31 lines 329-353) As one of the three main sections of the article, applications and clinical relevance were recapitulated and elaborated in the results (P30 lines 281-311, P31 lines 329-353), discussion (P33 lines 483-495), and conclusion (P33 lines 526-528).

Point 9: Limitations should be provided

Response 9: Thanks for the reviewer's insightful comment. We previously mixed up the limitations of research related to 4D virtual patients with that of the present systematic review. We have revised this in the discussion section and added more analysis of the limitations in this paper.

DISCUSSION: There are some limitations to this systematic review. First, the included studies were either non-randomized experimental studies or case reports. Few studies showed low overall risk. Therefore, the scientific level of clinical evidence is lacking. Appropriate statistical methods are needed to evaluate non-randomized experimental studies. The case reports lacked demographic characteristics, medical histories, and current clinical conditions. Second, the total sample size was 78, which needs further expansion. Third, there was considerable variation in subjects, interventions, outcomes, study design, and statistical methods across the included studies. Due to high heterogeneity, we did not conduct a meta-analysis but only qualitatively discussed the technique and application. Fourth, a few included articles were a series of related studies done by the same authors, which may increase the bias of the results. Although all criteria are met, such cases should be avoided in future studies. Finally, the number of included snowball papers was more than that obtained by electronic searches. There were no suitable articles in the weekly literature tracking. Thus, our search strategy needs to be improved. (P33 lines 496-509)”

Point 10: The conclusion should be rewritten based on the results; it is unclear.

Response 10: Thank you for pointing out this problem in our manuscript. We have restructured the conclusion in the revised manuscript. The first three conclusions are mainly aimed at three main sections, and the second two point out the limitations and future research directions. (P33 lines 517-533)

Point 11: Extensive editing of English language and style required.

Response 11: Thank you for your advice. We have used an alternative service to edit the English language and style. The confirmation certificate has been sent to the Editorial Office.

Reviewer 2 Report

The authors submitted a manuscript entitled “Four-dimensional superimposition techniques to compose dental dynamic virtual patients: a systematic review”. The aim was to summarize the current scientific knowledge in the dental dynamic virtual patient field to guide subsequent related research.

The Introduction is well-written and offers enough information regarding the topic.

The Materials and Methods section is complex and divided adequately into subchapters for a better understanding.

The Results section offers information regarding the included studies, as well as generously presented outcomes.

The Discussion and Conclusions sections are well designed.

I recommend the publication of this manuscript.

Best regards!

Author Response

Point: The authors submitted a manuscript entitled “Four-dimensional superimposition techniques to compose dental dynamic virtual patients: a systematic review”. The aim was to summarize the current scientific knowledge in the dental dynamic virtual patient field to guide subsequent related research.

The Introduction is well-written and offers enough information regarding the topic.

The Materials and Methods section is complex and divided adequately into subchapters for a better understanding.

The Results section offers information regarding the included studies, as well as generously presented outcomes.

The Discussion and Conclusions sections are well designed.

I recommend the publication of this manuscript.

Response: We thank the reviewer for reading our paper carefully and giving the above positive comments. We will continue maintaining the good parts and correcting the shortcomings in the subsequent research. Thank you again for your help!

Reviewer 3 Report

Dear Authors,

the idea of "four-dimensional virtual patien" review is interesting and deserves attention.The search was performed correctly, selecting a good number of scientific articles. The data is about to be collected and catalogued.

The discussions are good and the current situation has been highlighted.

To improve reading, I propose to insert graphs, diagrams, histograms to make the results more understandable. Tables provide a lot of information, but are sometimes hard to read

Author Response

Point 1: The idea of "four-dimensional virtual patient" review is interesting and deserves attention. The search was performed correctly, selecting a good number of scientific articles. The data is about to be collected and catalogued.

The discussions are good and the current situation has been highlighted.

Response 1: We are grateful for your comments on the manuscript and will make further efforts in the follow-up research.

Point 2: To improve reading, I propose to insert graphs, diagrams, and histograms to make the results more understandable. Tables provide a lot of information but are sometimes hard to read

Response 2: We appreciate your valuable suggestion. Indeed, our table is somewhat complicated because the 4D virtual patient is a complex technology combining multiple data. We have added Figure 2a and Figure 2b to illustrate better the percentage of different risk levels in the revised manuscript. The graph will provide a more visual representation of the risk of bias in the included articles. However, we have not inserted more diagrams, due to the lack of the software and hardware and related technologies required in the fusion process.

Reviewer 4 Report

I would like to see more discussion related with the application of these models in the context of numerical simulations (using for example the finite element method, in Abaqus or Ansys, for example. Also, just a suggestion, but many complex numerical models are also meshed with the software Hypermesh).

The manuscript would benefice with the inclusion of some images. 

Author Response

Point 1: I would like to see more discussion related with the application of these models in the context of numerical simulations (using for example the finite element method, in Abaqus or Ansys, for example. Also, just a suggestion, but many complex numerical models are also meshed with the software Hypermesh).

Response 1: Thank you for your rigorous consideration. We have added relevant content using Ansys and FORGE as examples in the results section 3.4.2 and discussed the relationship of these models and numerical models, Abaqus and Hypermesh in 4.2.2 Software programs. In addition, we have added the clinical information and applications included in this FE model in 3.4.3. However, since the included articles in this paper require more than two information acquisition methods to build dynamic virtual patients, the 4D virtual patient domain currently mentions only two articles for FE model analysis. As the reviewer said, the software Hypermesh meshed many complex numerical models, so we briefly described its characteristics in our discussion. Of course, our follow-up study will go deeper into the finite element method, contributing to the accuracy analysis mentioned in our future research directions.

RESULTS: “FE modeling and analysis software can convert 4D virtual models created by other software into numerical models and analyze dynamic/static components. The FE software commonly used for 4D virtual models is ANSYS (ANSYS Inc., PA, USA) [38]/FORGE (Transvalor, GLpre 2005) [39]. Processing software such as AMIRA (Vis-age Imaging, Inc, CA, USA) are often used to obtain surface and volume meshes. The volumes of the anatomical components were input to FE software and meshed as the element; subsequently, the mesh quality and nodes' quality were verified. The accuracy of a FE model is determined using geometric models. (P28 lines 252-259)”

“FE methods can simulate dynamic masticatory models, including muscle forces to the teeth, to determine the magnitude and direction of the bite force. High-resolution FE models analyzed the stress distribution and symmetry of TMJ and the boundary conditions of mandibular during the closure process. Further studies are expected to enable the prediction of various stress loads on the TMJ disc in the context of mandibular trauma, surgery, or dysfunction. (P31 lines348-353)”

DISCUSSION: “The 4D virtual model ensures that a comprehensive model contains the required data, while the FE analysis software chunks the complex model into simple units connected by nodes, facilitating simple algorithms for the analysis and interpretation of complex data. Although few FE software programs analyzed 4D models, several FE-related programs were used to analyze 3D models. For example, Hypermesh (Altair, Troy, MI) is an important preprocessing software, and Abaqus (Abaqus Inc., Providence, USA) is a common FE analysis software that interprets geometric models [45,46]. (P32 lines 422-428)”

Point 2: The manuscript would benefice from the inclusion of some images.

Response 2: Thank you so much for your suggestions. Indeed, the information needed to be integrated by 4D virtual patients and related technologies is very complex. Including some images will increase the article's readability and better explain the content. However, in addition to collecting patients' information from different sources, the construction of 4D virtual patients also needs to rely on complex software and hardware structures. We can gather relevant information, but we still have no software, hardware, or related technologies required in the fusion process. We aspire to make up for this deficiency with multiple tables. We also added Figure 2a and Figure 2b to illustrate better the percentage of different risk levels in the revised manuscript. Therefore, we seek the review’s tolerance and understanding. Many thanks for your kind help!

Round 2

Reviewer 1 Report

The paper has been significantly improved. 

Congratulations on your work!

Reviewer 4 Report

The authors have improved the manuscript.